# THINK SMALL, ACT BIG: PRIMITIVE-LEVEL SKILL PROMPT LEARNING FOR LIFELONG ROBOT MANIPULATION

## ABSTRACT

The general-purpose robots need to continuously acquire new skills in life-long spans without revisiting past experiences, known as Rehearsal-free Lifelong Learning, which remains significantly challenging. Recent advances learn a separate adapter along pretrained policy for each new skill to address catastrophic forgetting problem, ignoring the shared knowledge between old skills and new ones. To tackle these issues, we propose Primitive-level Skill Prompt Learning (PSPL), to achieve lifelong robot manipulation via reusable and extensible primitives. Within our two stage learning scheme, we first learn a set of prefix skill prompts to extract shared knowledge through multi-skills pre-training stage, where motion-aware skill prompts are learned to capture semantic and motion shared primitives across different skills. Secondly, when acquiring new skills in lifelong span, new prefix skill prompts are added and learned via cross-attention between prefix prompts of old skills, boosting the new skills learning via shared knowledge transfer. For evaluation, we construct a large-scale skill dataset and conduct extensive experiments in both simulation and real-world tasks, demonstrating PSPL's superior performance over state-of-the-art methods. Code and dataset will be released upon acceptance.

## 1 INTRODUCTION

Learning continuously without forgetting is an essential aspect of intelligence. As humans, we can effortlessly acquire and retain a vast repository of skills throughout our lives, all without explicitly revisiting past experiences. However, unlike humans, robotic agents often struggle with severe *catastrophic forgetting*, where learning new skills interferes with what was learned before. To alleviate this issue, previous approaches rely on storing and replaying previous data to maintain prior knowledge (Rolnick et al. (2019); Sodhani et al. (2020)), but this can be impractical in the real world due to memory limitations or privacy concerns. Beyond these methods, we direct our attention to a specific problem known as **R**ehearsal-**f**ree **L**ifelong **L**earning (**RfLL**). In this setting, agents must learn from a continuous stream of expert data without employing memory mechanisms to revisit past demonstrations.

To deal with RfLL, some work attempts to leverage regularization or dynamic architecture to achieve more efficient knowledge transfer compared to rehearsal-based counterparts Kirkpatrick et al. (2017); Zenke et al. (2017); Li & Hoiem

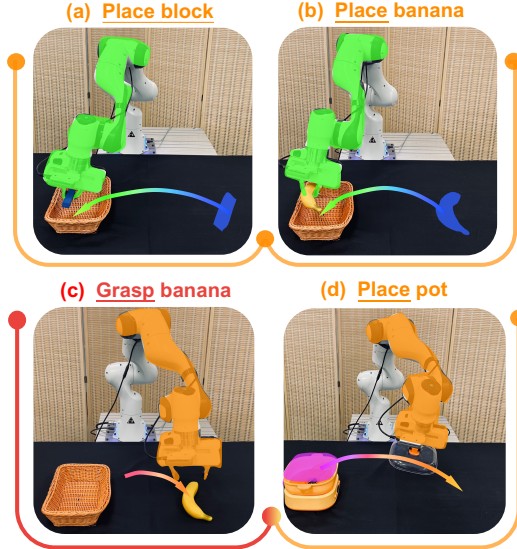

**(a) Place block**  **(b) Place banana**

**(c) Grasp banana**  **(d) Place pot**

Figure 1: Optical flow captures primitive-level motion patterns, revealing latent shared knowledge between semantically similar skills $(a, b)$ and distinct skills $(c, d)$.

(2017). These methods, primarily based on penalizing parameter changes and compartmentalizing model components, often struggle with poor performance as indirect knowledge-retrain strategies, especially scaling to the complex vision-based manipulations domain Aljundi et al. (2018); Serra et al. (2018). More recently, Liu et al. (2023) demonstrated the potential of using Low-Rank Adaptation (LoRA Hu et al. (2021)) and adapts skill-specific LoRA for each new skills, allowing for efficient parameter updates without interfering with previously learned skills. As shown in Fig. 1, skills like "Grasp banana" and "Place pot", while semantically different, may share common underlying motion primitives. Recognizing and leveraging these shared primitives is crucial for effective knowledge transfer and lifelong learning across diverse robotic skills.

To use these shared primitives in robotic manipulation, skill-based learning methods learn a set of primitive skills and reuse them for the acquisition of new skills Yin et al. (2023); Mandlekar et al. (2020); Xu et al. (2018b). These methods decompose complex robotic tasks into fundamental reusable skills, according to the hierarchical nature of human skill acquisition Kroemer et al. (2021); Peters & Schaal (2008), which requires sophisticated skill discovery and decomposition algorithms. The key advantage lies in the potential for knowledge transfer and scalability: as robots acquire a repertoire of primitive skills, they can combine and reuse them to tackle novel tasks Kober et al. (2013); Gao et al. (2022). More recently, LOTUS Wan et al. (2024) have attempted to integrate skill-based learning with lifelong robotic learning. However, it still requires experience replay to develop its skill library, posing substantial memory challenges as the number of tasks increases. Our research reveals that while existing methods employ various techniques for skill discovery and knowledge sharing, they have not fully explored how to effectively utilize shared knowledge for learning new skills.

In this paper, we propose Primitive-level Skill Prompt Learning (PSPL) for lifelong robot manipulation, a novel two-stage framework that transfers the knowledge across skills via reusable and extensible primitives. Our framework first learns a set of shared skill prompts to model shared knowledge through primitive-level multi-skills pre-training. Specifically, we introduce motion-aware skill prompt learning that adopt a text-flow query mechanism to capture semantic and motion shared primitives across skills. For individual skill learning, skill-specific motion-aware skill prompt is represented by weighted-sum of shared skill prompts and prepended into the keys and values of multi-head self-attention layers of diffusion transformer-based policy. In this way, the primitive-level shared knowledge learned and stored into the shared skill prompts. For new skill learning lifelong span, we add new prefix skill prompts into previous learned shared skill prompts, and learn them together with new skill demonstrations via cross-attention between old and new skill prompts. This intuitively enables knowledge transfer between old and new skills, without redundant new parameters and complex skill decomposition. To evaluate PSPL, we construct a large-scale skill dataset and conduct extensive experiments in both simulation and real-world tasks, demonstrating significant performance improvements over state-of-the-art methods. Our contributions are as follows:

- We propose Primitive-level Skill Prompt Learning (PSPL), tailored for achieving lifelong robot manipulation via reusable and extensible primitives.
- Motion-aware skill prompts and text-flow query mechanism are designed to capture shared semantic and motion knowledge between multiple skills and effectively transfer them to new skill acquisition.
- We construct a large-scale skill dataset and conduct extensive experiments in both simulated and real-world environments, demonstrating significant performance improvements over state-of-the-art methods in lifelong robotic manipulation.

## 2 RELATED WORK

**Lifelong Learning.** Lifelong learning for decision-making aims to develop an agent that can continuously learn and adapt to new tasks from a stream of data while retaining previous knowledge to avoid catastrophic forgetting Parisi et al. (2019); Lesort et al. (2020); Khetarpal et al. (2020). Prior **rehearsal-based** approaches involve storing and replaying past experiences to maintain the learned knowledge Rolnick et al. (2019); Shin et al. (2017); van de Ven et al. (2020). However, as the number of tasks increases, the memory requirements grow significantly, limiting their scalability for robot manipulation. Alternatively, another line of **rehearsal-free** work attempts to leverage regularization or dynamic architecture to achieve more efficient knowledge transfer compared to

rehearsal-based counterparts Kirkpatrick et al. (2017); Zenke et al. (2017); Li & Hoiem (2017). These methods, primarily based on penalizing parameter changes and compartmentalizing model components, often struggle with poor performance as indirect knowledge-retrain strategies, especially scaling to the complex vision-based manipulations domain Aljundi et al. (2018); Serra et al. (2018). Most recently, inspired by the advancements of parameter-efficient fine-tuning in language domains, TAIL Liu et al. (2023) with LoRA Hu et al. (2021) obtain state-of-the-art performance with a few trainable parameters in lifelong learning scenarios. However, TAIL requires maintaining specific parameters for each task and does not leverage learned knowledge to boost novel skill acquirement, making it inefficient in the real world.

**Skill-based Imitation Learning.** Skill-based imitation learning focuses on leveraging temporally abstract representations from sensory-motor data (termed *skills*) and learning a skill-conditional policy to accelerate the imitation process for some long-horizon manipulation tasks. A series of research segment expert demonstrations into sub-trajectories to learn these skill representations, employing unsupervised strategy Shankar et al. (2020); Abi-Farraj et al. (2020); Sharma et al. (2022) or relying on auxiliary supervised information Kipf et al. (2019); Lynch et al. (2020); Xu et al. (2018a). Additional work has shown promise for embedding fixed-length sub-trajectories without supervision through generative models such as variational auto-encoder Pertsch et al. (2020); Wang et al. (2021); Köhler et al. (2020) or diffusion model Janner et al. (2022); Chi et al. (2023); Xu et al. (2023). Furthermore, LISA Garg et al. (2022) incorporates skill learning with language instructions by sampling multiple skills per trajectory and uniquely integrating language conditioning. In this work, we adopt a similar setting due to its relative simplicity and scalability.

## 3 PROBLEM FORMULATION

Within our multi-skill pre-training, we consider a set of robot tasks $C = \{T_j\}_{j=1}^{J}$. For each task $j$, we have $N$ expert demonstrations $\{\tau_{j,i}\}_{i=1}^{N}$, where each demonstration $\tau_{j,i}$ is a sequence of state-action pairs. We formulate robot imitation learning as an action sequence prediction problem, aiming to minimize the error in future actions conditioned on historical states. The standard behavioral cloning loss is used to optimize the agent's policy $\pi$ over these demonstrations:

$$\hat{\theta} = \min_{\theta} \sum_{k=1}^{K} \mathbb{E}_{s_t, a_t \sim \mathcal{D}_k} \left[ \sum_{t=0}^{l_k} \mathcal{L} \left( \pi(a|s_t, T_k; \theta), a_k^t \right) \right]. \tag{1}$$

where $L$ is a supervised action prediction loss (e.g., mean squared error or negative log likelihood), $l_k$ is the length of demonstrations for task $T_k$, and $\theta$ refers to the learnable parameters of the network.

In lifelong learning span, we leverages the pre-trained model from the first stage, which not only showcases the model's scalability but also demonstrates the reusability of multitask pre-training in benefiting subsequent lifelong learning. Our objective remains to incrementally learn new skills while retaining performance on previously learned ones. The pre-trained agent continues to encounter a sequence of tasks, denoted as $T_1, ..., T_K$. For each task $T_k$, the agent receives $N$ demonstrations $D_k = \tau_k^1, ..., \tau_k^N$. A key characteristic of this stage, relevant to equation 1, is that after learning task $T_k$, the agent cannot access additional data from preceding tasks. In this context, $\mathcal{D}_k$ only contains data from the current task, and $s_t$ should be interpreted as $s_{\leq t}$. This constraint creates a rehearsal-free lifelong learning scenario, emphasizing the importance of transferring knowledge across tasks without risking catastrophic forgetting.

## 4 METHOD

### 4.1 OVERVIEW

The overview of our method is shown in Fig. 2. Given an input demonstration stream $\{D_i\}_{i=1}^{J}$ and a skill description T, we aim to leverage human demonstrations and task information to learn a set of reusable and extensible primitive-level skill prompts. In Sec. 4.2, we introduce motion-aware prompting to capture semantic and motion shared primitives across different skills, combining optical flow with task-conditional semantic information. Then, a two-stage training scheme is presented in Secs. 4.2 and 4.3, where we first learn prefix skill prompts to model shared knowledge through

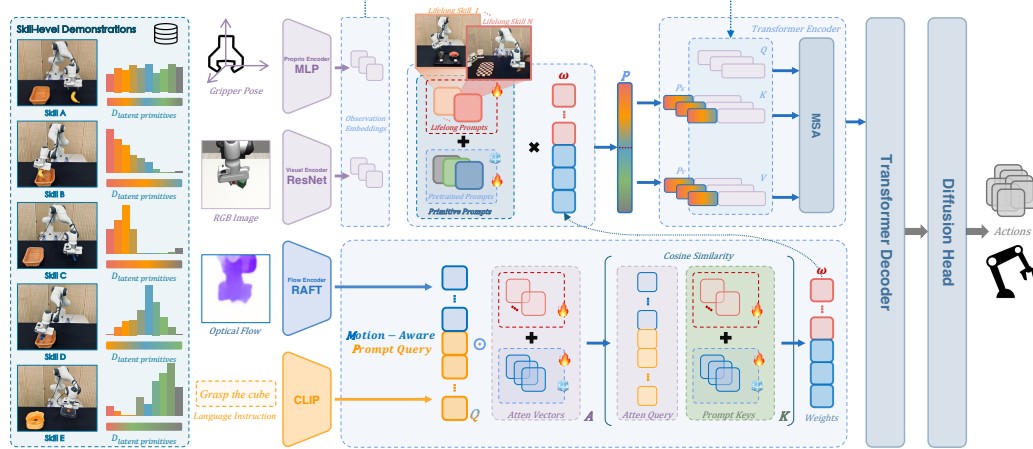

Figure 2: **The overview of Primitive-level Skill Prompt Learning (PSPL)**. In pre-training stage, given a large-scale dataset with numerous primitive-level skill demonstrations, the input consist of proprioception, image observation, optical flow and language instruction and a set of shared primitive skill prompts are queried via motion-aware query module to obtained a weighted-sum skill-specific prefix prompt, which is preprend to each layer of diffusion-transformer policy. For new skill acquisition with expert demonstrations, two new shared skill prompts are added and optimized with pretrained shared primitive skill prompts, following the same input/output flow as the pre-training.

multi-skills pre-training, followed by a lifelong learning phase that adds and learns new prefix skill prompts via cross-attention between old and new skill prompts. Finally, our method iteratively optimizes the skill representation by minimizing the reconstruction loss between observed demonstrations and generated motions, enabling primitive-level knowledge transfer across different skills and finally implementing lifelong skill acquisition.

## 4.2 PRIMITIVE-LEVEL SKILL PROMPT LEARNING

As shown in Fig. 2, in the first stage of our method, we utilize a diffusion transformer policy with our constructed skill dataset to perform multi-skill pre-training.

Specifically, we apply prefix-prompt learning to the diffusion transformer policy, instead of augmenting the input tokens, prepending prompts to the keys and values of the MSA layers, with distinct prompting parameters for each layer. We define our prompt parameter as $p \in \mathbb{R}^{L_p \times D}$, where $L_p$ represents the prompt length and $D$ denotes the embedding dimension. In a typical MSA layer with input $h \in \mathbb{R}^{L \times D}$, the query, key, and value are represented as $h_Q$, $h_K$, and $h_V$ respectively. The layer's output is computed as follows:

$$\text{MSA}(h_Q, h_K, h_V) = \text{Concat}(h_1, \ldots, h_m)W^O$$

$$\text{where } h_i = \text{Attention}\left(h_Q W_i^Q, h_K W_i^K, h_V W_i^V\right)$$

where $W^O$, $W_i^Q$, $W_i^K$, and $W_i^V$ are projection matrices, and $m$ denotes the number of attention heads. Our approach involves splitting the prompt $p$ into $\{p^K, p^V\} \in \mathbb{R}^{(L_p/2) \times D}$ and prepending these to $h^K$ and $h^V$ using the prefix-prompt method:

$$f_{P-T}(\mathbf{p}, \mathbf{h}) = \text{MSA}(h_Q, [\mathbf{p}_K; h_K], [\mathbf{p}_V; h_V])$$

However, there is a key limitation of approaches that heavily rely on high-level representations such as skill IDs or semantic information, which often face challenges in facilitating mutual improvement between tasks that are not semantically similar, potentially overlooking the rich temporal and motion information inherent in robotic actions. For example, while effective for knowledge transfer between semantically similar tasks like "grasp cube" and "grasp mug", these methods fall short in capturing shared primitives across semantically distinct but motion-related tasks. This limitation can result in sub-optimal knowledge transfer between seemingly unrelated tasks like "*grasp mug*" and "*place banana*", which, despite their semantic differences, may share common underlying primitives.

To address these limitations and capture semantic and motion shared primitives across different skills, we propose Motion-Aware Prompting (MAP). MAP combines optical flow with task-conditional semantic information, allowing us to capture and leverage common primitives across seemingly disparate tasks. Specifically, motion-aware optical flow information provides a rich representation of motion dynamics within the scene, capturing the essential kinematic properties of primitive actions. This motion-centric approach allows us to identify and learn common movement patterns across tasks, even when the high-level semantics differ. For instance, while "grasp cube" and "place mug" may seem semantically unrelated, they both involve the primitive of arm lowering. To capture these motion dynamics, we employ the Recurrent All-Pairs Field Transforms (RAFT) Teed & Deng (2020) model for optical flow estimation. In RAFT, the optical flow is computed iteratively:

$$f_{k+1} = f_k + \Delta f_k \tag{2}$$

where $f_k$ is the flow estimate at iteration $k$, and $\Delta f_k$ is the flow update computed as:

$$\Delta f_k, h_{k+1} = \text{GRU}(C(f_k), h_k) \tag{3}$$

Here, $C$ is a correlation volume, $h_k$ is a hidden state, and GRU is a gated recurrent unit. Optical flow effectively captures these shared motion primitives, enabling more granular knowledge transfer. Secondly, optical flow offers a degree of invariance to appearance changes, focusing instead on the underlying motion structure. This property is particularly valuable in robotics, where the same primitive-level manipulation might be performed on objects with vastly different visual characteristics.

$$I(x, y, t) = I(x + u\Delta t, y + v\Delta t, t + \Delta t) \tag{4}$$

where $I$ is the image intensity, $(u, v)$ is the optical flow vector, and $\Delta t$ is the time step. This allows optical flow to capture motion information while being relatively insensitive to the specific appearance of the scene. Concurrently, we embed conditional descriptions of tasks into a shared latent space using a pre-trained CLIP model. This allows us to leverage rich semantic understanding, providing a powerful representation of task semantics. By combining optical flow features with these task-conditional semantic embeddings, our Motion-Aware Prompting (MAP) achieves a dual purpose. We can represent this as:

$$\text{MAP}(T, F) = f_{\text{prompt}}(E_{\text{CLIP}}(T), \Phi(F)) \tag{5}$$

where $T$ is the task description, $F$ is the optical flow from RAFT, $E_{\text{CLIP}}(T)$ is the CLIP-based semantic embedding function, $\Phi(F)$ is a flow feature extraction function, and $f_{\text{prompt}}$ is a learned function that combines semantic and motion information. The CLIP-based semantic embedding ensures task-specificity, guiding the model towards relevant skills, while the flow feature enables fine-grained decomposition of skills into primitives. This approach enables our model to learn and transfer knowledge at the primitive level, thereby facilitating mutual improvement and lifelong expansion across diverse skills.

### 4.3 LIFELONG SKILL ACQUISITION

Parameter-efficient methods have shown remarkable success in mitigating catastrophic forgetting. However, current state-of-the-art approaches exhibits limitations in expanding learning capacity across tasks. They learn a single adapter for each new task, failing to leverage shared knowledge across different tasks. Therefore, we propose a novel lifelong skill acquisition method that during lifelong span, new prefix skill prompts are added and learned via cross-attention between prefix prompts of old skills, achieving helpful shared knowledge transfer from old skills to new ones.

Specifically, we introduce a new dimension to our learning capacity: a set of prompt components. Our method combines these components through weighted summation, forming a decomposed prompt that is subsequently fed into the corresponding MSA layer. This enables us to expand our prompting capacity to arbitrary depths while maintaining a fixed prompt length. Notably, when prompting for new tasks in lifelong learning contexts, our method reuses previously acquired knowledge from past tasks, rather than initializing a new task prompt from scratch. Formally, we replace the learnable prompt parameter p with a weighted summation over the prompt components:

$$p = \sum_m \alpha_m P_m \tag{6}$$

Here, $P \in \mathbb{R}^{M \times D}$ represents our set of prompt components, where M denotes the length of this set, introducing an additional axis of capacity. The critical aspect of this formulation is determining the appropriate weighting vector ($\alpha$) for each task.

To achieve dynamic prompt generation, we propose an innovative approach that computes the weight vector $\alpha$ based on the similarity between a primitive-based query $\theta(x)$ and a set of keys associated with the prompt components. This method allows for the production of primitive-based prompts without relying on the fixed task index. Specifically, the weighting vector is derived from the cosine similarity between the query and a set of keys:

$$\alpha = \gamma(q(\mathbf{x}), \mathbf{K}) = \{\gamma(q(\mathbf{x}), \mathbf{K}_1), \gamma(q(\mathbf{x}), \mathbf{K}_2), \ldots, \gamma(q(\mathbf{x}), \mathbf{K}_M)\} \quad (7)$$

where $K \in \mathbb{R}^{M \times D}$ represents keys corresponding to the prompt components. This formulation ensures that each prompt component $P_m$ contributes to the final prompt p in proportion to the similarity between the query q(x) and its corresponding key $K_m$.

The challenge inherent in this prompt-query matching lies in its similarity to high-dimensional clustering, a notoriously difficult problem. To address this issue, the authors introduce an attention mechanism to the key-query matching process. Each $P_m$ is paired with both a key $K_m$ and an attention vector $A_m$. This addition enables the query to focus on specific features within the high-dimensional query q(x) output, potentially capturing more primitive-based features while disregarding less relevant information. The implementation involves a straightforward feature-selection attention scheme. An element-wise multiplication between the query vector and the attention vector produces an attended query, which is then used for key-similarity matching. The refined approach to generating the weighting vector is expressed as:

$$\alpha = \gamma(q(\mathbf{x}) \odot \mathbf{A}, \mathbf{K}) = \gamma(q(\mathbf{x}) \odot \mathbf{A}_1, \mathbf{K}_1), \ldots, \gamma(q(\mathbf{x}) \odot \mathbf{A}_M, \mathbf{K}_M) \quad (8)$$

Here, $A \in \mathbb{R}^{D \times M}$ comprises learnable parameters (attention vectors) corresponding to the prompt components, and ($\odot$) denotes the Hadamard (element-wise) product. Notably, these attention vectors function as learnable feature weightings rather than input-conditioned modules.

---

**Algorithm 1** PSPL: Primitive-level Skill Prompt Learning

---

**Require:** Visual demonstrations $\{D_i\}_{i=1}^{J}$, Skill descriptions $T$
**Ensure:** Learned primitive-level skill prompts
1: Initialize $p \in \mathbb{R}^{L_p \times D}$                                        ▷ Initialize prefix skill prompts
2: **for** each skill $j$ in $\{1, \ldots, J\}$ **do**
3:     $f_{k+1} = f_k + \Delta f_k$                                  ▷ Compute optical flow using RAFT
4:     $\text{MAP}(T, F) = f_{\text{prompt}}(E_{\text{CLIP}}(T), \Phi(F))$            ▷ Motion-Aware Prompting
5:     $f_{P-T}(p, h) = \text{MSA}(h_Q, [p_K; h_K], [p_V; h_V])$      ▷ Apply prefix-prompt learning
6:     Compute diffusion loss $\mathcal{L}$                          ▷ Using diffusion transformer policy
7:     Update $p$ and model parameters to minimize $\mathcal{L}$
8: **end for**
9: **for** each new skill $k$ **do**
10:     Initialize $P \in \mathbb{R}^{M \times D}$                            ▷ Initialize new prompt components
11:     Compute $\text{MAP}_k$                                           ▷ Compute MAP for new skill
12:     $\alpha = \gamma(q(x) \odot A, K)$                        ▷ Compute attention-based weighting
13:     $p = \sum_m \alpha_m P_m$                                           ▷ Generate new prompt
14:     Compute diffusion loss $\mathcal{L}$ for new skill          ▷ Using diffusion transformer policy
15:     Update $p$ and model parameters to minimize $\mathcal{L}$
16:     Add $p$ to existing prompts                                        ▷ Expand prompt set
17: **end for**
18: **return** Learned prompts

---

# 5 EXPERIMENTS

## 5.1 EXPERIMENTAL SETUP

**Simulation tasks.** We conduct our simulation experiments using a large-scale skill dataset that we constructed based on MimicGen and LIBERO. In our skill dataset, each skill is associated with its

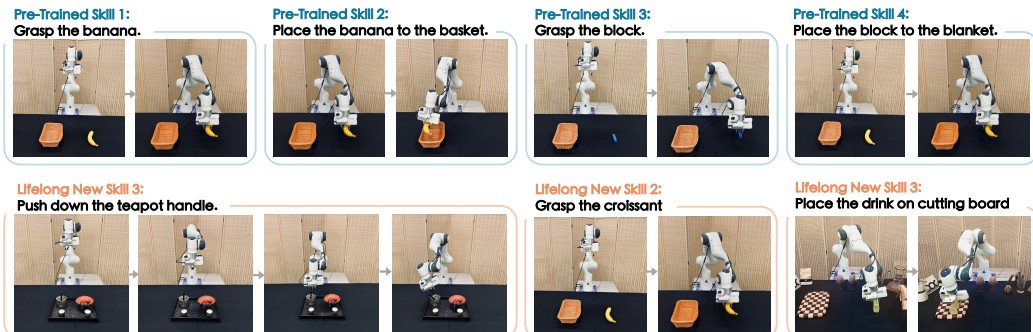

Figure 3: **Real-world robot setting.** We proposed 9 real-world skills, 4 of which are used in the pre-training stage and 5 in the lifelong stage, covering a variety of action spaces such as grasp, place, push, and a variety of different objects and distributions.

own natural language description. For example, a skill might be described as "Grasp the mug" or "Open the drawer". As shown in fig. 4, our dataset incorporates 24 skills from MimicGen, each containing 1K human demonstrations and with broad initial state distributions, effectively showing the generalization for multitask evaluation. We also include tasks from LIBERO, a lifelong robotic manipulation benchmark. Specifically, we utilize LIBERO-Goal, which focuses on the same scene with different goals. From LIBERO-Goal, we extract 11 skills, each comprising 50 human demonstrations. By building our large-scale skill dataset, we ensure a comprehensive range of robotic manipulation scenarios, enabling our policy on diverse and challenging tasks.

**Real-world experiments.** The real-robot experiments are conducted on the Franka Panda robotic arm. As shown in fig. 3, we perform multitask pre-training on four distinct skills, each comprising 200 human demonstrations with broad initial state distributions. To evaluate our policy's ability for lifelong learning, we conduct training and validation on four additional skill tasks. The objects involved in these tasks, such as banana, block, and various utensils, are randomly placed to assess position generalization. All metrics are evaluated with 10 independent runs for each skill, ensuring robust performance assessment across different initial conditions and task variations.

**Evaluation Metrics.** Following Liu et al. (2023), we employ Forward Transfer Weight (FWT) and Backward Transfer Weight (BWT) to evaluate the performance of lifelong learning. FWT is computed by the maximum success rate our policy can achieve when adapting to a new task. We denote FWT at task $k$ as $F_k$. Meanwhile, BWT measures the success rate increase on previous tasks. Specifically, when adapting to the $k$-th task, we first record the best FWT model on this task and then evaluate this model on all previous $k-1$ tasks, obtaining success rate $S_i$, $1 \leq i \leq k-1$. Then we compute the success rate difference between the new model and the best FWT of the previous $k-1$ tasks and then average among them to obtain the BWT metric:

$$B = \frac{1}{k-1} \sum_{i=1}^{k-1} (S_i - F_i), \tag{9}$$

For both FWT and BWT metrics, higher values indicate better performance in terms of knowledge transfer and retention across tasks.

## 5.2 MULTI-SKILL PRE-TRAINING

As shown in Table 3, our PSPL achieves the highest success rates across all pre-training tasks in the LIBERO-GOAL environment. Compared to the MOE, our method improves the average success rate across all tasks by 17%. We further evaluate our method's ability to learn generalizable cross-skill information in real-world scenarios. Table 3 presents the results of real-world experiments, where our policy consistently outperforms existing approaches. These results validate our method's effectiveness in both simulated and real-world environments.

## 5.3 LIFELONG LEARNING

For lifelong learning tasks, we conducted a comparative analysis of our method against traditional sequential learning approaches, experience replay-based methods, and task-specific LoRA. As il-

Figure 4: **Illustration of our primitive-level skill dataset.** The primitive-level skill dataset is constructed based on MimicGen benchmark with diverse action spaces and scene variations.

lustrated in Tables 1 and 3 , our method demonstrated superior performance in simulated environments, achieving state-of-the-art performance in both FWT and BWT metrics. Furthermore, Table 3 presents evidence that in real-world scenarios, our approach not only facilitates the acquisition of cross-skill premitives during the pre-training phase but also effectively leverages this premitives in the new skill learning stage. Notably, our method surpasses existing approaches without requiring access to replay experiences.

| Task | Conventional Methods | | | | Adapter-based Methods | |
|------|--------|--------|--------|--------|--------|--------|
| | Sequential | | ER | | LoRA | PSPL (Ours) |
| | FWT ↑ | BWT ↑ | FWT ↑ | BWT ↑ | FWT ↑ | FWT ↑ |
| Task 1 | 0.87 ± 0.07 | - | 0.79 ± 0.12 | - | **0.89** ± 0.02 | 0.88 ± 0.00 |
| Task 2 | 0.73 ± 0.07 | -0.57 ± 0.08 | 0.71 ± 0.07 | -0.23 ± 0.08 | **0.79 ± 0.01** | 0.75 ± 0.12 |
| Task 3 | 0.79 ± 0.04 | -0.48 ± 0.12 | 0.67 ± 0.07 | -0.37 ± 0.11 | 0.81 ± 0.07 | **0.83 ± 0.03** |
| Task 4 | 0.77 ± 0.03 | -0.62 ± 0.17 | 0.64 ± 0.07 | -0.44 ± 0.19 | 0.78 ± 0.00 | **0.79 ± 0.02** |
| Task 5 | 0.49 ± 0.07 | -0.69 ± 0.24 | 0.35 ± 0.14 | -0.57 ± 0.23 | **0.62 ± 0.12** | 0.60 ± 0.09 |
| Task 6 | 0.64 ± 0.12 | -0.66 ± 0.24 | 0.52 ± 0.19 | -0.61 ± 0.23 | 0.61 ± 0.12 | **0.73 ± 0.14** |
| Task 7 | 0.32 ± 0.05 | -0.69 ± 0.18 | 0.11 ± 0.00 | -0.58 ± 0.24 | 0.43 ± 0.26 | **0.54 ± 0.11** |
| Average | 0.65 ± 0.06 | -0.56 ± 0.16 | 0.61 ± 0.09 | -0.46 ± 0.18 | 0.78 ± 0.09 | **0.83 ± 0.03** |

Table 1: **Lifelong Performances with MimicGen.** PSPL achieved the best success rate in both multi-skill pre-training and lifelong learning, as well as demonstrating superior lifelong learning capabilities.

| Task | Methods | | |
|------|---------|-----|------|
| | Diffusion-Transformer | MOE | Ours |
| | Multi-Skill Policy Pre-Training | | |
| Pretrain Task 1 | 0.60 ± 0.05 | 0.82 ± 0.04 | **0.99 ± 0.03** |
| Pretrain Task 2 | 0.25 ± 0.06 | 0.78 ± 0.05 | **0.62 ± 0.02** |
| Average | 0.42 ± 0.09 | 0.73 ± 0.08 | **0.84 ± 0.05** |
| | Lifelong Learning | | |
| Task | Sequential | ER | Ours |
| Lifelong Task 1 | 0.60 ± 0.08 | 0.65 ± 0.07 | **0.72 ± 0.04** |
| Lifelong Task 2 | 0.55 ± 0.09 | 0.58 ± 0.08 | **0.68 ± 0.05** |
| Lifelong Task 3 | 0.50 ± 0.10 | 0.52 ± 0.09 | **0.63 ± 0.06** |
| Average | 0.55 ± 0.09 | 0.58 ± 0.08 | **0.68 ± 0.05** |

Table 2: **Performances with real-world robot tasks.** PSPL achieved the best success rate in both multi-skill pre-training and lifelong learning, as well as demonstrating superior lifelong learning capabilities.

## 5.4 ABLATION STUDIES

**Effect of Motion-Aware Prompt Query** To validate the effectiveness of our motion-aware text-flow query, we visualize the weight distributions when using only text as the query and when using our text-flow query. As shown in the figure 6, if only text is used as the prompt query, the weight responses will only exhibit similarities in semantically related tasks, and within a single task, the

| Task | Methods | | |
|------|---------|------|-------------|
| | Diff-T | MOE | PSPL (Ours) |
| Multi-Skill Pre-Training | | | |
| Pretrain Task 1 | $0.79 \pm 0.05$ | $0.83 \pm 0.04$ | $\mathbf{0.85 \pm 0.03}$ |
| Pretrain Task 2 | $0.83 \pm 0.11$ | $0.85 \pm 0.03$ | $\mathbf{0.86 \pm 0.02}$ |
| Pretrain Task 3 | $0.84 \pm 0.07$ | $0.86 \pm 0.08$ | $\mathbf{0.86 \pm 0.01}$ |
| Pretrain Task 4 | $0.63 \pm 0.08$ | $0.74 \pm 0.07$ | $\mathbf{0.80 \pm 0.03}$ |
| Average | $0.55 \pm 0.09$ | $0.58 \pm 0.08$ | $\mathbf{0.68 \pm 0.05}$ |
| Lifelong Learning | | | |
| Task | Sequential | ER | Ours |
| Lifelong Task 1 | $0.77 \pm 0.08$ | $0.73 \pm 0.04$ | $\mathbf{0.78 \pm 0.04}$ |
| Lifelong Task 2 | $0.65 \pm 0.03$ | $0.61 \pm 0.12$ | $\mathbf{0.68 \pm 0.09}$ |
| Lifelong Task 3 | $\mathbf{0.74 \pm 0.11}$ | $0.62 \pm 0.08$ | $0.71 \pm 0.06$ |
| Average | $0.72 \pm 0.04$ | $0.65 \pm 0.03$ | $\mathbf{0.73 \pm 0.03}$ |

Table 3: **Performances with LIBERO-GOAL.** When dealing with different tasks in the same scene, PSPL still achieves the best performance.

Figure 5: **Simulation setting of LIBERO-GOAL.**

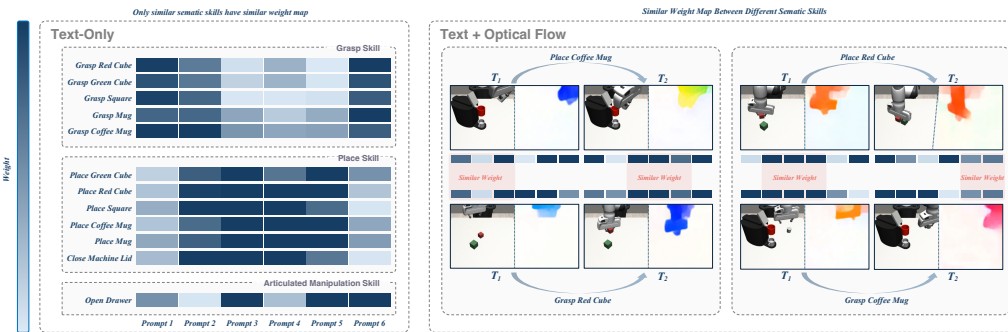

Figure 6: **Impact of Motion-Aware Prompt Query on Prompt Weights.** This figure illustrates the weight distributions when using only text as the query (left) and when using our text-flow query (right). When only text is used as the prompt query, the weight responses exhibit similarities only in semantically related tasks. In contrast, our text-flow query enables the policy to have similar weight responses even in semantically different skills, allowing different skills to learn primitives in the latent space.

weights remain the same at each time step. In contrast, our text-flow query enables the policy to have similar weight responses even in semantically different skills, allowing different skills to learn primitives in the latent space.

**Effect of Skill Prompt Count**  We conducted a comprehensive investigation into the optimal selection of prompt count during the multi-skill learning. As various skills undergo joint optimization, primitives are encoded and stored within prompts. For any specific task, only a subset of prompts responds and matches to extract relevant prior knowledge, while unmatched prompts may introduce noise. Consequently, as illustrated in Figure 7, an increase in the number of prompts does not necessarily correlate with improved performance. Simultaneously, an insufficient number of prompts may fail to encompass all primitives, underscoring the importance of achieving an appropriate balance in prompt count.

**Effect of Primitive Skill Prompt**  As illustrated in Figure 7, significant performance degradation is observed when learning new skills under two conditions: (1) when prompt learning of primitives is omitted during the pre-training phase, or (2) when pre-trained prompts are not utilized in the acquisition of new skills. These findings substantiate the effectiveness of our proposed prompt mechanism in extracting common knowledge from pre-trained skills. Moreover, they demonstrate the mechanism's capacity to repurpose this knowledge during the lifelong learning phase, thereby enhancing the performance of newly acquired skills.

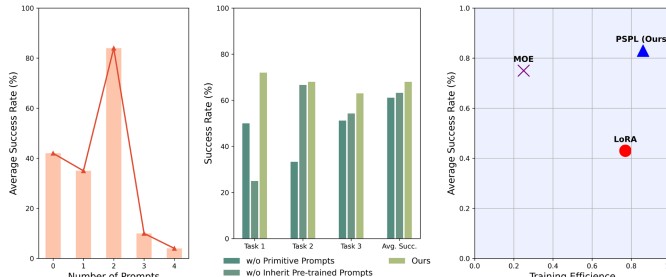

Figure 7: **Illustration of ablation studies.** We conducted ablation analysis on the effect of skill prompt count, the effect of primitive skill prompt, and comparisons with MoE and LoRA.

### 5.5 DISCUSSION ON OUR METHOD V.S. LORA AND MOE

Recently, some studies have explored the effectiveness of LoRA Liu et al. (2023) and MOE Wang et al. (2024) in enhancing lifelong robot learning. However, as illustrated in Figure 7, our experiments demonstrate that although MOE excels in terms of average success rate, its training speed is slower due to the additional computational overhead introduced by its gating network and multiple expert networks. MOE's training time is approximately twice that of LoRA and our proposed method. LoRA, on the other hand, emerges as the frontrunner in terms of training speed, while its overall performance falls short of its competitors. Notably, our method achieves performance surpassing that of MOE while maintaining comparable training speed. This balance of efficiency and efficacy enables our approach to effectively combine the strengths of LoRA and MOE, facilitating faster skill knowledge acquisition while preserving high performance.

## 6 CONCLUSION AND LIMITATION

In this work, we present Primitive-level Skill Prompt Learning for lifelong robotic skill learning. Motion-aware skill prompts and text flow query mechanism are proposed to learn reusable and extensible primitive-level knowledge across multiple skills and achieve superior results in multi-task policy learning. Moreover, for new skill acquisition, new skill propmts are easily added and learned for knowledge transfer between old and new skills, without redundant new parameters and complex skill decomposition. Finally, we construct a large-scale primitive-level skill dataset and demonstrate the superior perform of our method in multi-task policy learning and lifelong new skill acquisition.

**Limitations:** Our method requires the pre-processed primitive-level skill dataset for pre-training stage, which is difficult to do for various human daily tasks. Moreover, motion-aware prompts relies on optical flow estimators, which is unstable in lighting variation interactive environments. Future work will focus on scaling our method to more daily tasks and extending the method to handle more challenging lighting scenarios.

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
