# OpenReview forum: "Think Small, Act Big: Primitive-level Skill Prompt Learning for Lifelong Robot Manipulation"
_ICLR.cc/2025/Conference — ICLR 2025 Conference Withdrawn Submission_

### Official Review · Reviewer_JgRJ · 2024-11-01

**Soundness:** 3
**Presentation:** 2
**Contribution:** 2
**Rating:** 5
**Confidence:** 2

**Summary:**

This work addresses the challenges of rehearsal-free lifelong learning, where robots must learn new skills continuously without revisiting previous data, by leveraging reusable and extensible primitives that capture shared motion and semantic knowledge across various tasks. The approach utilizes a two-stage training process: first, it learns a set of motion-aware skill prompts during a multi-skill pre-training phase, which uses a combination of optical flow and semantic embeddings to identify shared knowledge at the primitive level. In the lifelong learning phase, PSPL expands its skill repertoire by adding new skill prompts that interact with pre-existing ones through cross-attention. Evaluation on a large-scale dataset and in real-world experiments demonstrates that PSPL achieves higher success rates in both pre-trained and newly acquired skills.

**Strengths:**

1. Preventing catastrophic forgetting by adopting skill-based learning even without a
memory mechanism.
2. Using optical flow acquired from RAFT and CLIP embedding together results in
discovering fine-grained motion primitives.

**Weaknesses:**

1. Requires a priori knowledge of skills (primitive-level skill demonstrations). All skill demonstrations require natural language descriptions. The authors made a dataset consisting of human demonstrations and natural language descriptions. Constructing such datasets is known to be expensive and the cost of it might surpass the cost of just using some memory mechanisms. Also, this specific requirement makes extending the work to other settings difficult (as the authors also pointed out in the last section).
2. The skills learned by the methods are simple and plain enough that they can be easily decomposed into sharable primitive behaviors. There is no evidence that can be found in the paper that the proposed method can be generalized into learning more complex and composed skills represented by longer prompts. For example, the skill dataset illustrated in [Figure 4] only deals with already decomposed descriptions like ‘open the drawer’, ‘grasp the mug’, and ‘place the mug’. The descriptions could also be thought as a single prompt ‘put the mug in the drawer’, but the dataset does not include the sort of prompts. This limitation poses generalizability and robustness issues to the proposed method.

**Questions:**

1. There are no explanations of baseline methods, which makes it difficult for the readers to understand specifically what are the improvements and implications of the proposed method compared to the baselines.

Minor:
[54-56] The sentence is exactly same with [109-111]. (...?)
[142] Typo: we leverages → we leverage
[179] Typo: to obtained → to obtain
[180] Typo: preprend → prepended
[292] There are no brackets on the RHS of the equation.
[Figure 3] Images attached to lifelong new skill 2 (grasp the croissant) are wrong. The robot is grasping a banana, not the croissant.

---

### Official Review · Reviewer_4QGD · 2024-11-02

**Soundness:** 2
**Presentation:** 1
**Contribution:** 2
**Rating:** 3
**Confidence:** 4

**Summary:**

This paper focuses on learning primitive skills for manipulation tasks and then using them for continual learning. There are two parts of the overall pipeline in this paper. First, a multi-skill diffusion policy is learned. This policy uses not only language conditioning but also motion conditioning via optical flow.  During this stage a set of prompts are learned for each skill which uses these two conditioning information. For new skill learning, new skill prompts are added but learned from cross attention with existing skill prompts in a weighted similarity based setting. The experimental result seem to suggest that the proposed method works well against a few baselines.

**Strengths:**

The paper aims to tackle an important problem of continual learning for manipulation tasks. The algorithm seems to have novel components but there seem to be too many moving pieces (more on this below). The experimental results suggest that the proposed approach works well (however I think the baselines can be significantly improved).

**Weaknesses:**

The overall paper can sometimes be very hard to read and understand. Some statements in the paper seem to be quite general (and sweeping) without much context. For instance line 256, “Parameter-efficient methods have shown remarkable success in mitigating catastrophic forgetting. …” There should ideally be citations to such claims. As far as I know, adapters do forget less but they still forget and unfortunately they also learn less (Biderman et al.).

One of the big assumptions made in this paper is that language labels for the task may not provide useful shareable information between tasks. This is considered as the main motivation for the first part of the paper. While this could be a problem, this problem could be alleviated by providing denser and more richer language labels. For instance, in Figure 1) instead of simply saying “place pot”, we can instead say “Keep the pot grasped, reach close to <target object> and then place on object”.  While optical flow can provide motion information which can be useful, it is unclear if denser and more descriptive language labels definitely do not provide this information. Since the labels used in this work are too disparate, I am not sure this claim can be made. I think it would be much more interesting to see if denser language labels do not provide this information.

Another advantage of dense language instructions would be that they can be easy to provide with narration settings (as popularized with some recent works).

How well does optical flow work for more complex tasks? Also, instead of optical flow some works have looked at keypoint tracking or object tracking to use motion information (e.g. [1] and [2]). This paper should compare against these works to show case that optical flow is indeed a better motion representation as compared to other prior works.

Continual learning: For  continual learning baselines I know the paper seems to emphasize rehearsal free learning, but it’s unclear to me why this setting is especially important. In tTable 2, the paper suggests that ER based baseline is barely better than sequential. This is extremely surprising to me since memory based approaches perform very strongly in continual learning settings. What is the intuition that ER completely fails here? Also, there are many other methods that have been built on top of ER, were any of these tried? Also, how was the memory size selected for ER based approaches. Also, why were other rehearsal free approaches such as EWC (or many others) not tried?

For using adapters for continual learning: The language and to some extent vision community has already focused a lot on continual learning (e.g. see Ke et al., Wu et al.) I think these methods should be cited and compared against.

Why are the task prompts applied to each layer as prefixes? This seems like an uncommon choice among state-of-the-art methods, is there a particular reason for this? There should be some experiment/ablation for this. Also how does the splitting happens for prefix prompts i.e. from p to p_k and p_v.

How are the prompt tokens (P_m) decided?  How are these prompt tokens related to prompts in section 4.2?

How much does the assumption that new skills being learned should indeed be related with existing skills affect the results? Would the approach fail if during continual learning phase we end up learning a skill that is not related to any existing skill? This is less of a problem with foundation models, but definitely a problem in the robotics setting with limited skills)

**Questions:**

please see above

---

### Official Review · Reviewer_rgNU · 2024-11-03

**Soundness:** 2
**Presentation:** 3
**Contribution:** 2
**Rating:** 5
**Confidence:** 3

**Summary:**

This paper studies the problem of continuous skill learning for robot manipulation. The framework is a transformer-based architecture that utilizes optical flow, RGB image, robot state information, and language instruction as input. The proposed method first pre-trains the model with large-scale robot skill dataset. While adapting to new tasks, a few demonstrations are used as prompts to fuse with the pre-trained prompts to learn new skills. Experiments are performed in both simulation and real world to validate the idea.

**Strengths:**

The problem of continuous robot skill learning is fundamental, the problem setting is well-established in the paper.

It seems like the method outperforms other baselines from evaluation perspective.

The paper is well-structured and easily read.

**Weaknesses:**

The method is conditioned on estimations from pre-trained models (e.g., optical flow), which may introduce errors that propagate and compound in downstream prediction tasks.

The comparison with baselines is insufficient, more baselines (e.g., EWC[1], PACKNET[2]) need to be included.

Some experiment details are missing or inaccurate, e.g., in line 341, MimicGen is machine-generated dataset, not human demonstrations; the description of MimicGen tasks should be more detailed.

The reference is inadequate, some relevant prior works on continuous/lifelong robot learning need to be discussed (i.e., [3][4]).


[1] Overcoming catastrophic forgetting in neural networks, NeurIPS’17.
[2] Packnet: Adding multiple tasks to a single network by iterative pruning, CVPR’18.
[3] Search-Based Task Planning with Learned Skill Effect Models for Lifelong Robotic Manipulation, ICRA’22.
[4] LEAGUE: Guided Skill Learning and Abstraction for Long-Horizon Manipulation, RA-L'23.

**Questions:**

What is the performance of the method on other LIBERO benchmarks (i.e., LIBERO-Spatial, LIBERO-Object, and LIBERO-100)?

What are the failure modes of the proposed method?

---

### Official Review · Reviewer_Bcw9 · 2024-11-05

**Soundness:** 2
**Presentation:** 3
**Contribution:** 2
**Rating:** 5
**Confidence:** 3

**Summary:**

This paper presents a framework for learning lifelong learning of robotic manipulation policies. The framework is a transformer-based model that jointly processes image, optimal flow, proprioception and language information and outputs diffusion policy. Experiments are conducted on the simulation environments of LIBERO-Goal and real-world settings.

**Strengths:**

- This paper deals with an important and interesting problem.
- The paper is well-written and easy to understand.
- The experiment are conducted on both simulation and real-world settings and the results outperform the baselines.

**Weaknesses:**

- Not sure if optical flow from RAFT, visual encoding from ResNet and language encoding from CLIP are all needed. If you have CLIP (a vision-language model), why do you still need ResNet to encode visual features? What if optical flow is removed? Does it still work?
- Not sure about the design choice of the model architecture. Why are lifelong prompts and primitive prompts are summed up before product with motion-aware prompts to compute $P$? Since you used transformation decoder at a later stage, why not just concatenate all of them?
- Not sure how are "primitives" defined? Can the "primitives" be learned? Can it be visualized?
- The colors are very confusing in Figure 2.
- Did the wrong images are place in Figure 3 for "Grasp the croissant"?

**Questions:**

- What is MSA layer? It is not defined.
- How are "primitives" defined? Can the "primitives" be learned? Can it be visualized?
- If you have CLIP (a vision-language model), why do you still need ResNet to encode visual features? What if optical flow is removed? Does it still work?
- In line 11 of Algorithm 1, what is the result of $MAP_k$? Where is the result used?

---

### Note · Authors · 2024-11-13

I have read and agree with the venue's withdrawal policy on behalf of myself and my co-authors.